# ARIES: A Corpus of Scientific Paper Edits Made in Response to Peer Reviews

## Abstract

Revising scientific papers based on peer feedback is a challenging task that requires not only deep scientific knowledge and reasoning, but also the ability to recognize the implicit requests in high-level feedback and to find an effective way to update the manuscript in response. We introduce this task for large language models and release ARIES, a dataset of review comments and their corresponding paper edits, to enable training and evaluating models. We study two versions of the task: comment-edit alignment and edit generation, and evaluate several baselines, including GPT-4. We find that models struggle even to identify the edits that correspond to a comment, especially in cases where the comment is phrased in an indirect way or where the edit addresses the spirit of a comment but not the precise request. When tasked with generating edits, GPT-4 often succeeds in addressing comments on a surface level, but it rigidly follows the wording of the feedback rather than the underlying intent, and includes fewer technical details than human-written edits. We hope that our formalization, dataset, and analysis will form a foundation for future work in this area.

## 1 Introduction

With remarkable recent advances in natural language processing capabilities, there has been increasing interest in systems that can reason about scientific content and help accelerate scholarly work (Hope et al., 2023). This includes assisting in tasks such as literature review (Luu et al., 2021; Li et al., 2022), reading (Chang et al., 2023), writing (Fok and Weld, 2023; Shen et al., 2023; Mahlow, 2023; Gmeiner and Yildirim, 2023) and hypothesis formation (Kang et al., 2022).

In this paper we focus on a task that encapsulates multiple challenges in reasoning about scientific text: revising papers in response to peer review feedback. This task provides a testbed for evaluating NLP systems on important and understudied capabilities needed for effective scientific assistants—performing the task requires a deep understanding of the full text of a scientific paper, and the ability to infer the intent behind technical human feedback and act upon it (revise the paper).

Feedback on paper drafts, whether from co-authors, readers, or reviewers, can be challenging to interpret and address because it often includes complex critiques of a paper's substance and can be phrased in an indirect way. For example, consider a reviewer who wants authors to use a more realistic dataset in their evaluation. This could be expressed in a variety of ways; it could be stated as a direct request (*"Apply the method to a realistic dataset"*), or more indirectly as a criticism (*"The evaluation is only on a synthetic dataset"*) or as a question (*"Is the current dataset truly representative of the real-world?"*). Similarly, an author editing the manuscript in response has several options: they could comply with the request, or clarify that no realistic datasets are publicly available, or even argue that the reviewer is mistaken and add a justification of their dataset's realism.

In this work, we evaluate whether large language models (LLMs) possess the reasoning abilities required to model the relationship between feedback and edits. We release **ARIES** (**A**ligned, **R**eview-**I**nformed **E**dits of **S**cientific Papers), a real-world dataset of computer science paper drafts, the corresponding reviewer feedback, and the author responses and revisions that address the feedback.[1]

Using this dataset, we formulate two novel tasks, shown in Figure 1: **comment-edit alignment**, in which a model must determine which review comments made about a paper correspond to each of

---

[1]The dataset and code will be released upon publication.

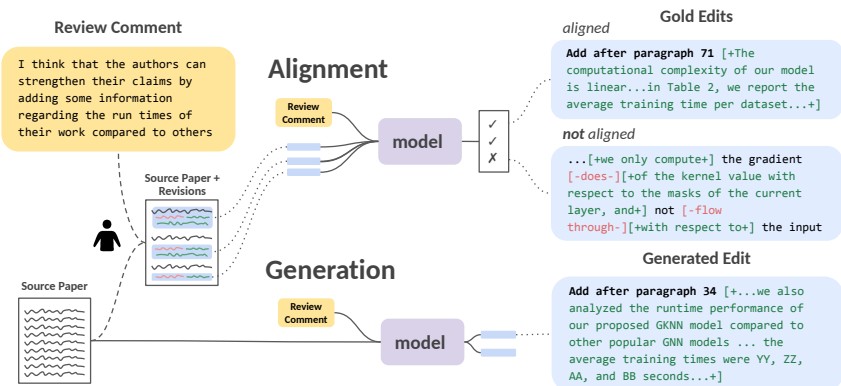

Figure 1: Overview of our tasks. In comment-edit alignment, a model is given a review comment and set of candidate edits derived from a source paper and a revised target paper, and it must align the comment to the edit(s) that are associated with it. In edit generation, a model is given a review comment and a source paper and must generate an edit that addresses the comment, possibly using placeholders for missing information.

the edits made after the feedback, and **edit generation**, in which a model must generate edits directly from a given reviewer comment and paper text.

In addition to serving as challenging testbeds for LLM evaluation, these tasks have the potential to advance assisted reading and writing applications. Automatic alignment could enable tools that allow readers to quickly find parts of a document that address particular questions or comments (ter Hoeve et al., 2020; Dasigi et al., 2021) or that help authors, reviewers, and area chairs more easily track revisions. Edit generation could power collaborative writing tools that allow authors to rapidly iterate on their manuscripts in response to feedback.

We evaluate ten baseline methods and find that the alignment task is challenging for existing models, including even large models such as GPT-4, and that comments and edits with indirect relationships are especially difficult. For the generation task, we find that GPT-4 does produce edits that are coherent and on-topic on a surface level, but fails to model the underlying intent; unlike real authors, it almost never makes edits that suggest the feedback is mistaken, often paraphrases the feedback rather than tightly integrating edits into the paper, and tends to include less technical detail.

In summary, our contributions are as follows:

- We propose the novel tasks of (1) aligning high-level draft feedback to specific edits and (2) generating revisions for scientific papers given reviewer feedback (section 3).

- We construct ARIES, a real-world dataset containing 196 human-labeled review comments matched to their corresponding paper edits, as well as 3.9K reviewer comments automatically matched to edits using author responses from OpenReview, with 92% precision (section 4).

- We evaluate a wide range of baseline methods on our comment-edit alignment task, finding that it is challenging even for modern LLMs. The best model (GPT-4) achieves only 27.0 micro-F1 compared to human performance of 70.7 (section 5).

- We conduct a thorough analysis of edit generation with GPT-4, detailing several systemic differences between generated and real edits, and suggest future work directions (section 6).

## 2 RELATED WORK

To our knowledge, our work is the first to study contentful edits conditioned on complex feedback in a highly technical domain (scientific papers). Previous work on edit modeling either focuses on stylistic and grammatical edits or incorporates no feedback or very different kinds of feedback—such as instructions or post-hoc descriptions of edits. Those settings don't present the same challenging

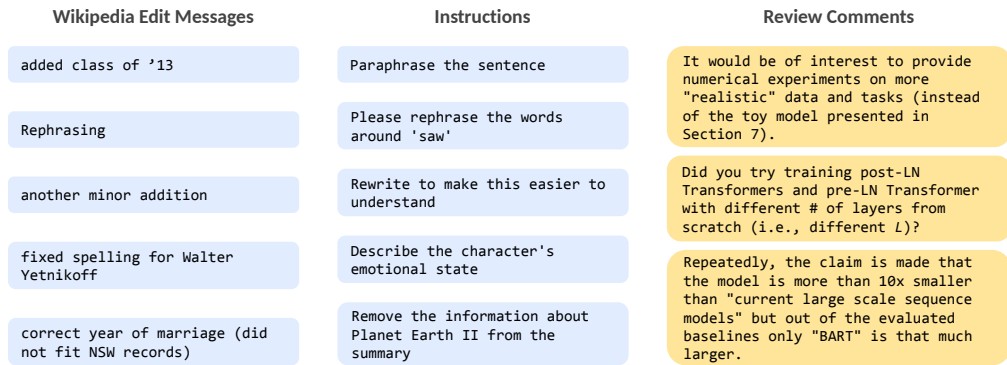

Figure 2: Representative examples of the kinds of conditioning information used to guide edits in our work (review comments) compared to previous work which considered Wikipedia edits (Faltings et al., 2021) and author-provided instructions (Ito et al., 2020; Yuan et al., 2022; Liu et al., 2022; Raheja et al., 2023). Review comments are longer and less direct, requiring more knowledge and reasoning to interpret.

reasoning requirements as our tasks. Figure 2 illustrates how the content and linguistic complexity of review comments differs substantially from that of the conditioning information used in past work.

**Style and Grammar Edits** Early work on edit modeling focused on grammatical error correction (GEC), which aims to identify and correct grammatically incorrect or misspelled text, and work in this area dates back several decades (Kukich, 1992; Wang et al., 2021). With the increase in language modeling capabilities in recent years, there has been progress in making more sophisticated edits such as rewriting a sentence to improve clarity, style, or structure (Malmi et al., 2019; Mallinson et al., 2022; Kim et al., 2022). However, these areas of research do not target the kinds of substantive revisions often made to papers in response to reviews, such as adding an entire sentence or paragraph to discuss a result or justify a design choice.

**Assisted Writing Systems** Several works develop writing assistants that incorporate human input to guide the edits. In some cases the human input is restricted to specific actions, such as marking words that the system should omit (Grangier and Auli, 2018) or selecting proposed edits to apply (Lee et al., 2022; Du et al., 2022a), while in other cases the user can provide a natural language instruction (Ito et al., 2020; Reif et al., 2022; Liu et al., 2022; Yuan et al., 2022; Raheja et al., 2023). However, the kinds of instructions found in these works are different from the draft feedback we investigate in that they are written by humans who know they are interacting with an automated system, resulting in more direct and specific instructions than the open-ended feedback that authors often receive for a draft.

Much of the previous research on edit modeling focuses on Wikipedia, using Wikipedia edit messages as a proxy for instructions when generating edits (Faltings et al., 2021; Schick et al., 2022; Reid and Neubig, 2022). Wikipedia edit messages are generally written post-hoc and provide varying levels of information about the content of the edit, often giving only a very vague summary like "add reference". In contrast, review comments generally provide enough information for a human to identify the content of the necessary edit.

ArgRewrite (Zhang et al., 2017; Kashefi et al., 2022) is a dataset of student essay revisions with teacher feedback, and contains some contentful comments, but the essays are very short (~500 words) compared to scientific papers (~5000 words) and the comments are not aligned to specific edits.

**Scientific Edits** Some work does explore scientific-domain edits, but these don't associate edits with reviewer comments and often focus on classification rather than generation. Jiang et al. (2022) and Du et al. (2022b) analyze and tag edit intentions on ArXiv papers but do not use feedback. Du et al. (2022a) develop a system for human-in-the-loop editing in several domains, including Wikipedia and Arxiv, but the feedback is limited to accepting/rejecting suggested edits. Mita et al. (2022) construct a dataset and evaluation framework for scientific document revision, including some

document-level revisions such as reordering sentences. Nonetheless, the aim is to improve writing quality rather than to alter the semantics of the text, and peer review comments are not used. Finally, Kuznetsov et al. (2022) identify edits between paper versions and separately align reviewer comments to referenced text in the source paper, but do not explore the connection between feedback and edits.

# 3 TASK DEFINITIONS

As shown in Figure 1, we consider two versions of the task of determining how a document should be edited to address a given piece of feedback: comment-edit alignment and edit generation. Both tasks express the differences between an original (source) document and revised (target) document as a list of *edits*, where each edit represents a specific change from source text at some location in the paper into new text in the target paper. Specifically, an edit consists of a paragraph in the source and its corresponding revised paragraph in the target, where either paragraph (but not both) can be null in the case of deletions or additions.

In the **comment-edit alignment** task, the goal is to identify the edit(s) that correspond to a given review comment. The input is a comment and a list of edits, which include both original and revised text. In our evaluation, we derive the list of input edits by using a paper's gold revisions, but they could consist of any candidate revisions. The output is a set of binary classifications over the list of edits, indicating whether each edit addresses the comment. Note that this results in a many-to-many mapping; one comment may result in several edits to the paper, and (less commonly in our data) multiple comments may be addressed by one edit.

In the **edit generation** task, the objective is to generate appropriate edits to a paper based on feedback. The input for this task consists of a comment and the original paper text. The output is the generated edit, which should address the reviewer's feedback and be coherent within the context of the paper.

# 4 DATASET CONSTRUCTION

Both the comment-edit alignment and edit generation tasks require a dataset with paper edits aligned to specific feedback comments. In this section, we provide an overview of our approach for collecting and annotating ARIES, a corpus of computer science paper revisions and reviews with both manual and synthetic annotations of comment-edit alignments. The following paragraphs provide a high-level overview of our dataset construction process, and the full details can be found in Appendix A.

First, we obtain a corpus of paper draft PDFs, their peer reviews, and revised drafts from OpenReview (subsection A.1) and manually identify token spans in reviews that represent actionable feedback (subsection A.3). We define *actionable feedback* as feedback that states or implies a specific request that could be addressed by an edit to the paper. Actionable feedback can be phrased in a wide variety of ways, including as questions or as implicitly negative remarks. However, a positive comment (*"The paper is sound and of certain interest"*) or one that simply summarizes the paper is *not* considered actionable for our purposes.

After identifying actionable feedback comments, we manually identify the (paragraph-level) edits that correspond to each comment to obtain a small (196 comments) but high-quality dataset for evaluating models. Cohen's $\kappa$ annotator agreement over a sample of this dataset is 0.8. We note that the mapping is many-to-many: some comments correspond to multiple edits, others do not correspond to any edit, and in some cases multiple comments correspond to the same edit. More details on this step can be found in subsection A.4.

To obtain a larger dataset (3.9k comments) suitable for training models, we develop a synthetic labeling approach to automatically extract comments and align them to edits using author responses (subsection A.5). The labeling algorithm is based on the observation that author responses often directly quote review comments and respond with text that is very similar to the corresponding edit text. These cases can be detected efficiently and reliably, resulting in a high-precision (>90%) but low-recall approach for identifying comment-edit pairs. Statistics of our final dataset are in Table 7.

## 5 COMMENT-EDIT ALIGNMENT

In this section, we evaluate models on the comment-edit alignment task using our constructed dataset. As described in section 3, the comment-edit alignment task is a binary classification task where the input is a review comment and a list of candidate edits, and the output is binary for each comment-edit pair, specifying whether the comment and edit are aligned. In model inputs, edits are textually represented using a "diff" format with additions and deletions enclosed in `[+ +]` and `[- -]` brackets.

For manually-annotated data, for a given comment, we consider all edits for the corresponding paper as candidate edits, labeled as positive if the edit was annotated as addressing the comment and negative otherwise. Given the low recall of the synthetic data (discussed in subsection A.5), we can only use the synthetic labels to produce positive comment-edit alignment pairs; thus, we pair comments with edits sampled from other documents as negative candidates. Additional details are provided in Appendix E.

### 5.1 MODELS

We consider four kinds of model architectures to align comments and edits: **Bi-encoders** (DeBERTaV3-large (He et al., 2021) and SPECTER2 (base) (Singh et al., 2022)), **Pairwise cross-encoders** (DeBERTaV3-large (He et al., 2021), LinkBERT (Yasunaga et al., 2022), and GPT-4 (OpenAI, 2023)) which score comment-edit pairs, **Multi-edit** cross-encoders (GPT-4) which consume all comments and edits for a paper at once (and output a list of alignments), and pairwise **Bag-of-words** (BM25 (Robertson and Zaragoza, 2009)). Specific details of these methods can be found in subsection E.1.

As an additional baseline, we apply BM25 using generated edits from GPT-4 (discussed in section 6) and refer to this as BM25-generated. As we show in section 6, GPT-generated edits are competitive with human edits in terms of the overall comprehensiveness with which they address comments, but they also differ substantially from human edits in style and content. The BM25-generated baseline serves as a way to empirically probe the similarity of the two kinds of edits.

Finally, as a weak baseline we evaluate a random classifier (using the proportion of positive examples in the test set as the positive probability), and as a strong baseline we evaluate how well an expert human annotator can perform on this task given the same inputs as the models. That is, the human is shown a comment and a full diff of the parsed source and target papers, but—unlike the annotators who labeled the task data—does not have access to author responses with which to identify unintuitive responses or to the PDFs with which to identify parsing errors.

### 5.2 RESULTS

Table 1 reports precision, recall, and F1 scores for models. The micro- scores are computed over all comment-edit pairs, while the macro- scores are macro-averaged by paper[2] to down-weight cases where a model incorrectly predicts many edits for one difficult comment. In addition to results over the full dataset, we also run experiments on just edits that add a full paragraph as addition-only F1 (AO-F1); this setting is easier because it does not require models to understand which tokens have been added, removed, or unchanged, and is a better fit for BM25, which cannot represent the differences between these tokens. Results are averaged over three trials with different random seeds for training. The prompt templates used for GPT-4 can be found in Appendix C.

We find that the task is challenging, with none of the models reaching human-level performance. GPT-4 methods are best, but interestingly it appears that giving GPT-4 a full chunk of the document at once (GPT-4 multi-edit) results in slightly worse performance than the pairwise approach, aside from an improvement in efficiency.

Across all methods, including human performance, we observe that macro-F1 is substantially higher than micro-F1, suggesting that some comments are especially error-prone. For example, 55% of GPT-4 multi-edit's errors correspond to just 20% of the comments. Nuanced comments on documents with many edits may lead to several incorrect predictions—*e.g.,* if they involve many small changes

---

[2]Implementation note: F1 is considered 100 for comments where the number of true alignments and predicted alignments are both zero. This can result in a bias favoring conservative models, which we discuss in Appendix F.

Table 1: Precision (P), Recall (R), and F1 of comment-edit alignment on test data. The micro-average is over all comment-edit pairs, while the macro-average is grouped by paper. Addition-Only F1 (AO-F1) is the F1 score when only addition-only edits are considered; due to budget constraints, this is the only feasible setting for pairwise cross-encoder GPT. Overall, GPT-4 methods are all much better than the smaller locally-trained models, but none reach human performance.

| Model | Micro | | | | Macro | | | |
|---|---|---|---|---|---|---|---|---|
| | AO-F1 | P | R | F1 | AO-F1 | P | R | F1 |
| BM25 | 13.3 | 12.2 | 10.5 | 11.3 | 48.0 | 41.7 | 26.4 | 20.9 |
| BM25-generated | 14.7 | 4.6 | 40.3 | 8.3 | 31.0 | 5.2 | 57.5 | 8.6 |
| Specter2 (no finetuning) | 14.0 | 8.1 | 14.4 | 10.3 | 42.3 | 22.2 | 29.0 | 13.0 |
| Specter2 bi-encoder | 19.6 | 17.0 | 29.3 | 21.5 | 40.2 | 34.6 | 38.1 | 22.6 |
| DeBERTa bi-encoder | 3.1 | 9.9 | 12.2 | 10.8 | 42.8 | 52.4 | 22.0 | 18.6 |
| LinkBERT cross-encoder | 2.8 | 10.1 | 28.4 | 14.4 | 41.0 | 14.7 | 40.8 | 12.9 |
| DeBERTa cross-encoder | 8.5 | 7.4 | 25.6 | 10.0 | 42.6 | 13.2 | 40.4 | 10.7 |
| GPT-4 cross-encoder 0-shot | 38.7 | - | - | - | 50.8 | - | - | - |
| GPT-4 cross-encoder 1-shot | 42.1 | - | - | - | 57.0 | - | - | - |
| GPT-4 multi-edit | 36.2 | 24.2 | 30.4 | 27.0 | 50.6 | 31.6 | 28.2 | 26.3 |
| Random | 5.5 | 1.5 | 1.7 | 1.6 | 20.2 | 10.2 | 17.6 | 4.9 |
| Human | 70.6 | 65.6 | 76.8 | 70.7 | 75.4 | 84.0 | 69.2 | 67.0 |

to technical details and equations—whereas other instances may be more straightforward. In the next section, we analyze specific failure modes that we observe.

## 5.3 FALSE POSITIVES

One author examined 50 randomly-sampled false positives of the best-performing model, GPT-4 multi-edit, and identified four common categories of mistakes that it makes. The categories and their frequencies are described in the following paragraphs. Note that the categories are partially overlapping, so the total exceeds 100%, and 10% of the errors did not fit clearly into any category.

**Too-Topical (40%)** In some cases, the model assigns a positive label to an edit that contains some words that are topically or lexically similar to the words in the review comment, but do not actually address the comment. In many cases, this happens even when the words are only part of the original text and were not added or deleted in the edit.

**Diff-ignorance (28%)** In some cases, a comment asks about something that is already present in the paper in some form—*e.g.,"add CIFAR10 experiments"* when there is already one CIFAR10 experiment in the paper, or asking to remove a misleading claim. The model sometimes aligns these comments to edits of paragraphs with preexisting (or deleted) content that is relevant to the comment, failing to correctly account for the add/delete markup.

**Over-Generation (28%)** This failure mode is unique to the multi-edit task format, in which models attempt to generate a full list of all comment-edit alignments for a paper in one pass. We observe some cases where GPT-4 outputs more consecutive edits in a list than it should; for example, if edits 17 and 18 are relevant to some comment, the model might add 19, 20, 21, 22 and so on. In rare cases, the list extends beyond the highest edit id in the input. Although it is difficult to precisely determine the factors that influence GPT-4's output, we hypothesize that GPT-4 may be suffering in part from exposure bias, and as it begins to generate a consecutive sequence it gets stuck in a loop and fails to stop at the correct place. This phenomenon has previously been studied in smaller models (Chiang and Chen, 2021), and may be occurring to a much lesser degree with GPT-4 as well.

**Bad Parsing (12%)** Some errors are simply the result of the PDF parser extracting text differently for different versions of a paper, causing text to appear edited when it was not. In some of these cases, the "edits" in question do look as though they partially address the comment, similar to the errors

in the "diff-ignorance" category, and the model erroneously (albeit reasonably) aligns to those edits without realizing they were already in the original paper.

## 5.4 FALSE NEGATIVES

We attempt to quantify how the explicitness of the relationship between a comment and edit affects alignment performance. We leverage two metrics: The first is a measure of **edit compliance**: Specifically, we annotate how directly an edit obeys a given comment on a 1-3 scale (1 being least compliant, 3 being most compliant). More details on the metric and compliance annotations are in section 6. The second is a measure of **comment directness**: how "direct" or "indirect" the comments are. A direct comment is one that indicates a clear action; this could be phrased in the negative, but still explicitly specifies what needs to be done (*"It is unfortunate that you didn't do experiments on imagenet"*). An indirect comment does not state the specific request, and is usually a statement of fact or observation that requires an understanding of linguistic and scientific norms to interpret (*"Only one dataset was used"*).

We measure the performance impact of indirectness and compliance on the multi-edit GPT-4 method in Table 2, and we find that both factors result in a substantial difference. GPT's micro-F1 is 30% lower on indirect comments compared to direct ones, and 24% lower when edits are non-compliant. These results suggest that GPT-4 struggles to understand complex comment-edit interactions and performs better on those with simple, direct relationships.

Table 2: Alignment micro-F1 for GPT and humans on direct/indirect comments and compliant/non-compliant edits. Note that the values are higher than in Table 1 because comments with no corresponding edits were not annotated. GPT and humans both do much worse with indirectly-phrased comments. GPT also struggles to match to non-compliant edits, whereas humans are unaffected.

|  | **Direct comment** | **Indirect comment** | **Compliance = 3** | **Compliance < 3** |
|---|---|---|---|---|
| GPT-4 | 40.4 | 28.2 | 39.5 | 30.1 |
| Human | 78.6 | 61.3 | 71.5 | 77.3 |

## 6 EDIT GENERATION

### 6.1 EXPERIMENTAL SETUP

Our goal is to understand the differences in style and content between the kinds of edits human authors write and those that models generate, which will provide insight into model behavior and point to directions for future improvements. However, we note that evaluating the *correctness* of generated edits is beyond the scope of our analysis, as it is both difficult to do fairly (models may lack information such as lab notes and raw data) and difficult to do correctly (robust judgements require a very deep expertise in a given paper).

We generate edits with GPT-4, which was the best model for comment-edit alignment and is known to be a powerful general-purpose generation model (OpenAI, 2023).

### 6.2 MANUAL ANALYSIS

We explore the differences between GPT-written and author-written edits more deeply with an analysis by two expert judges (authors of this paper, with multiple CS/ML publications) on 85 comments. The comments were divided between the two judges, except for 10 instances that were annotated by both in order to measure agreement. Each instance includes the original paper, the review comment, and both GPT's generated edits and the set of real edits that were made to the paper in response to the comment. The judges are aware of which edits are model-generated and which are real, as it would be impossible to conceal the stylistic differences; however, we do not believe this impacts our goal of understanding the trends between the two edit types, as the judges scored edits using several specific factors described in the following rubric. Examples of these factors can be found in Table 3:

Table 3: Examples of comment-edit pairs exhibiting each scored factor in the edit generation analysis (subsection 6.2). Edits marked with an asterisk (*) are generated by GPT, while the others are real. Text is ellipsized for brevity.

| Factor | Comment | Edit |
|---|---|---|
| Compliance=1 | ... Isn't this percentage too much? Can't we use, e.g., 5% of all nodes for training? | [+... our split of 80% -10% -10% is a standard split+] |
| Compliance=2 | ... there is a hyprameter in the radius decay, how it will affect the performance is crucial ... | [+... this learnable radius is not effective the in terms of an classification performance compared to that the predefined radius decay+] |
| Compliance=3 | the experimental setup requires significantly more details on the hardware ... | [+We conducted our experiments using NVIDIA Tesla V100 GPUs ...+]* |
| Promises | it would be interesting to know how the proposed method would work, for instance, for node classification (e.g., Cora, Citeseer) | [+... the performance of our method on node classification tasks is beyond the scope of this paper and is left as an interesting direction for future work.+]* |
| Paraphrases | ... it should be investigated ... with respect to more natural perturbations, e.g. noisy input, blurring, ... | [+... we also investigate their performance with respect to more natural perturbations, such as noisy input, blurring, ...+]* |
| Technical details | ... This does put into question whether the full closed loop model is actually useful in practice | [+... we evaluated the performance of a closed-loop N-CODE model ... Here, the control parameters are a matrix of dynamic weights, $\theta(t) \in \mathbb{R}^{m \times m}$ ...+] |

- **Compliance (1-3):** The edit might argue that the comment is irrelevant or infeasible to address (1), address the spirit of the comment but not specifically what was asked (2), or directly comply with the reviewer's advice (3).

- **Promises (true/false):** The edit promises to address part of the comment in future work or a future revision; we include cases where the model says it provides information elsewhere (e.g., in its Appendix) but does not give the corresponding edit for that section.

- **Paraphrases (true/false):** The edit reuses the wording from the comment itself.

- **Technical details (true/false):** The edit contains specific details or placeholders for details such as citations, mathematical expressions, or numerical results.

We note that the edit generation task is made technically impossible by the fact that some edits may require information that the model does not have, such as the results of additional experiments. We mitigate this by instructing the model to use placeholders or to hallucinate technical details that it does not know (details in Appendix C). In addition, for each comment we measure **answerability**: whether it can be addressed *without* placeholders or hallucinations. In other words, a perfect model should be able to address answerable comments using just the original paper and background knowledge.

Additionally, for each (GPT, real) edit pair, we evaluate which has greater **comprehensiveness** in addressing the reviewer's comment, as there are many cases where one edit is more thorough or goes beyond what the reviewer asked, even though both have the same compliance. This is not the same as correctness; instead, comprehensiveness measures how thoroughly an edit *attempts* to address a comment, possibly using placeholders or hallucinating unavailable information.

## 6.3 RESULTS

From an initial inspection of GPT's generated edits, we find that the model almost always produces coherent and on-topic edits. Table 4 shows that GPT-generated edits are competitive with human-authored edits in comprehensiveness, often being rated as more comprehensively addressing the given comment when sufficient information is available but doing worse for comments that require additional data to address. On average, GPT almost matches real edits in this regard.

Table 4: Fraction of cases where GPT or real edits were deemed more comprehensive than the other, broken down by answerability. Frequency is the fraction of all comments that fall into each category. Overall, GPT and are comparable to real edits, with GPT being better for comments that don't require additional data and real edits being better for those that do.

Table 5: Edit generation analysis. We report average Compliance and % of examples that include each of the other factors. We report Cohen's $\kappa$ for all factors on 10 instances and p-values using Wilcoxon's signed-rank test for Compliance and Fisher's exact test for others. GPT is more compliant, often paraphrases the comment, and tends to include fewer technical details than real edits.

|  | Ans. | Non-ans. | All |
| --- | --- | --- | --- |
| GPT | 31% | 19% | 25% |
| Real | 19% | 40% | 29% |
| Same | 50% | 42% | 46% |
| Frequency | 51% | 49% | 100% |

|  | GPT | Real | $\kappa$ | p |
| --- | --- | --- | --- | --- |
| Compliance | 2.9 | 2.6 | 0.6 | $10^{-4}$ |
| Promises | 21% | 6% | 1.0 | $10^{-2}$ |
| Paraphrases | 48% | 4% | 0.7 | $10^{-11}$ |
| Technical details | 38% | 53% | 0.7 | 0.06 |

However, we observe in Table 5 that the kinds of edits generated by GPT-4 are very different than those produced by real edits. The most striking difference we observe is the tendency for GPT-4 to paraphrase the comment when writing its edit. Qualitatively, we notice that GPT-4's edits are often written as though they are meant to be a standalone response, whereas the real edits are more tightly integrated into the context of the paper. In addition, real edits are more likely to use specific technical details as opposed to a high-level response, an effect which is understated in Table 5 due to the cases where both edits contain some technical details but one contains substantially more. To account for these cases, we additionally record relative technicality judgements ("more", "less", "same") for each (GPT, real) edit pair and find that the difference grows: the real edits are more technical in 38% of cases compared to only 12% for GPT (p=$10^{-3}$). Overall, the reduced technicality and the tendency to paraphrase may make GPT-4's edits preferable for those who simply want clear responses to their questions and feedback, but they also make edits less informative for the most engaged readers who care about technical details.

We also note that while most edits from both GPT-4 and humans follow the reviewer's specific instructions, human edits deviate from the reviewer's request more often: 94% of GPT-4 edits are highly compliant (compliance = 3), while only 68% of human edits are. The actual discrepancy in this factor may be even higher, as real authors often choose not to make an edit at all when they disagree with a comment, opting instead to discuss it on the OpenReview forum.

## 7 CONCLUSION AND FUTURE WORK

In this work, we have introduced the novel tasks of comment-edit alignment and edit generation for scientific paper revisions based on high-level draft feedback from reviewers. We have constructed and released a dataset containing pairs of computer science paper drafts with edits aligned at the paragraph level, along with their corresponding reviews and author responses. We hope the dataset will enable research on assisted writing technologies and serve as a challenging testbed for LLMs.

It is interesting that models (including GPT-4) do so poorly on the comment-edit alignment task despite GPT being able to generate plausible edits in the generation task. As our analysis shows, the kinds of edits produced by GPT can be very different from the real edits authors make to their papers, and the fact that GPT fails to recognize many of the real comment-edit pairs suggests that it may have gaps in its reasoning that would be interesting to explore further in future work. We hope that the insights from our analyses can help motivate and guide future studies.

A shortcoming of the generated GPT edits is their relative lack of technical details. However, this may be caused in part by their lack of access to information about detailed experimental results, code, and lab notes for the paper, which the authors have when doing their revisions. As a long-term goal, we believe that an ideal writing assistant would observe the entire research process and consume all relevant information when writing an edit; in some cases, this might even include suggesting additional experiments for humans to run. However, this requires further work both to create applications that can collect this information and to develop efficient methods to provide this information to large language models, which are currently limited in input size and expensive to run.

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

# A  DATASET CONSTRUCTION DETAILS

## A.1  COLLECTING PAPERS AND REVIEWS

We obtain papers, reviews, and author responses from computer science conferences on OpenReview.[3] For each paper, we use the latest PDF that was uploaded before the first review as the original version and the latest available PDF as the revised version. We omit papers that do not have a revised version uploaded after reviews were posted, resulting in a set of 6,501 paper records. We use Grobid (GRO, 2008–2023) and S2ORC (Lo et al., 2020) to parse the paper PDFs.

## A.2  DETECTING EDITS

We identify edits between the source and target papers by finding pairs of paragraphs with high bigram overlap.

To perform initial alignment of drafts, we create a mapping $m(j) \to i$ from a paragraph $t_j$ in the target document to paragraph $s_i$ in the source document. For each pair, we score the similarity as

$$\text{sim}(i, j) = \text{bigram}(s_i, t_j) - \frac{|m(j-1) + 1 - i|}{|S|}$$

where $|S|$ is the number of paragraphs in the source document and bigram() is the bigram overlap. The term $\frac{|m(j-1)+1-i|}{|S|}$ is used to encourage matching to a paragraph close to where the previous paragraph was matched to. We take the most similar match if $\text{sim}(i, j) > 10\%$, otherwise we consider it a new paragraph. Unmapped source paragraphs are considered to be deleted.

In some cases, PDF parsing errors cause a paragraph to be split differently in different drafts. E.g., the paragraph may be broken across a page boundary differently, so the parser thinks it is two paragraphs. To prevent this from resulting in spurious "edits", we post-process all matches to check if the similarity would become higher by merging $s_i$ or $t_j$ with an adjacent paragraph. If so, we merge them.

On average, a paper revision typically has 40% of its paragraphs unchanged, 14% "minor" edits (with less than 10 tokens changed, usually fixing typos or grammar), 14% "major" edits, 8% fully deleted paragraphs, and 23% fully new paragraphs.

## A.3  IDENTIFYING ACTIONABLE FEEDBACK

To create our manually-annotated evaluation data (196 instances), we first extract sentences from reviews which constitute actionable feedback. We define *actionable feedback* as feedback that states or implies a specific request that could be addressed by an edit to the paper. Actionable feedback can be phrased in a wide variety of ways, including as questions or as implicitly negative remarks. However, a positive comment (*"The paper is sound and of certain interest"*) or one that simply summarizes the paper is *not* considered actionable for our purposes.

Two annotators manually annotated 42 reviews to extract the token spans corresponding to actionable feedback (details in Appendix D), ultimately resulting in 196 comments. In some cases, a comment might only make sense in the context of some other sentence from the review. For example, in *"The paper is missing several things: (1) a definition of L, (2) ImageNet baseline, (3) ..."*, the phrase "ImageNet baseline" is only interpretable in the context of the top-level comment. Where this occurs (9% of comments), we annotate both the context and comment spans and concatenate them into a single comment.

Inter-annotator agreement was measured on a set of 10 reviews that were annotated by both annotators, with a total of 60 non-overlapping spans between the two annotators. We find that 88% of spans

---

[3]https://openreview.net

overlap between annotators, but due to differences in amounts of included context the token-level Jaccard overlap is 65%. In subsection B.1, we conduct further analysis on the types of actionable review comments in our extracted data.

### A.4   ALIGNING COMMENTS TO EDITS

The extracted actionable comments (subsection A.3) were mapped to their corresponding edits in the paper by an expert annotator (an author of this paper). For each comment, the annotator was given the original and revised paper PDFs and the list of edits and asked to identify which edits were made in response to the comment. As additional context, the annotator was given the responses authors made to the reviewers on the OpenReview forum to assist with finding all of the intended edits, as authors often state in their response where they made edits to address each point made by the reviewer. Agreement was calculated against a second annotator on a sample of 25 comments, obtaining a Cohen's $\kappa$ of 0.8.

In total, 78% of comments were addressed by the authors. However, 28% were addressed only in the author response and not with edits to the paper, and 7% were addressed in the paper but not visible in the parsed text (either because of a parsing error, or because the edit was purely visual, such as changing a figure), leaving 43% (85 comments) aligned to textual edits (the comments without edits are still included as challenging examples for our comment-edit alignment task). The aligned comments each correspond to 2.1 edits on average.

### A.5   CREATING SYNTHETIC DATA

To produce a large training set with high-quality comment-edit alignments, manual annotation is not feasible; each review takes approximately 30 minutes to fully process and requires annotators with extensive domain expertise, and our corpus contains 24k reviews. Thus, we automatically generate a large silver dataset of comment-edit alignments by leveraging the fact that authors often quote reviewer comments directly in author responses, and the edits that correspond to a comment are often highly similar to the author response text discussing the comment.

For synthetic comment-edit alignment data generation, we automatically identify the quoted review comments in author responses by searching for lines with a small edit distance to a contiguous span of review text (with a minimum length of 40 characters, to eliminate spurious matches). The corresponding response text for each comment is matched to edits with high textual overlap; we informally observe that edits with at least 25% bigram overlap to the response text almost always correspond to the quoted comment. Using this threshold, we link responses and edits to obtain a set of 3892 high-precision alignments from the training corpus.

Unlike the manually-annotated data, the synthetic data has low recall; applying the synthetic labeling algorithm to our hand-labeled data identifies only 2% of the matches. However, they have high precision: We manually checked 50 sampled alignments and found that 46 were correct. Furthermore, we find that the synthetically-aligned data has similar statistics to the manually-annotated data; see subsection B.3 for details.

## B   DATA ANALYSIS

In this section, we discuss observations about the kinds of edits and review comments we find in our dataset.

### B.1   COMMENTS

To explore the kinds of comments found in reviews, we asked annotators to categorize extracted comments in the manually-annotated data partition according to what kind of action the comment request from the author, using the following action classes:

- **Compare** a proposed method or resource to a baseline from prior work
- **Apply** a proposed method or theory to a different task or dataset
- **Use** a method from prior work to improve a proposed method

- **Define** a term or symbol
- **Discuss** a related paper
- **Report** an additional metric or analysis for an existing experiment or observation
- **Explain** a detail about the proposed method or finding
- **Remove** something, such as a confusing or misleading claim

Table 6: Rates at which different action classes occurred in comments and the frequency with which they were actually addressed by authors in their revisions.

| Action | Occurred | Addressed |
|---|---|---|
| Explain | 42% | 39% |
| Compare | 14% | 32% |
| Report | 10% | 39% |
| Remove | 8% | 50% |
| Apply | 8% | 33% |
| Use | 8% | 33% |
| Define | 7% | 47% |
| Discuss | 7% | 67% |

The results of our analysis are summarized in Table 6; 7% of comments did not clearly fit any category and were omitted from this analysis, leaving 182 comments. We observe that comments asking to compare with a new baseline, use a new component in a proposed method, or apply the same method to an additional dataset or setting were the least likely to be addressed in revisions. This is likely because those kinds of requests require (potentially substantial) additional experimental work to be done. Requests to define terms or discuss related work were the most commonly addressed, although the small number of comments in those categories means those estimates are likely to be high-variance.

### B.2   EDITS

Most (71%) edits made in response to review comments consist of solely adding a contiguous span of text. Many edits both add and delete text (34%), and very few (2%) consist of only deletions. On average, an edit adds 89 tokens and deletes 12.

### B.3   COMPARISON OF SYNTHETIC VS. MANUALLY-ANNOTATED DATA

We ask whether the synthetically aligned data (subsection A.5) results in different kinds of comments and edits than the manually-annotated data. Surprisingly, we find that the two sets of data have similar statistics for most of the properties we measure. The synthetic data has the same ratio of edits that add a full new paragraph as opposed to altering an existing paragraph (64% vs 65% new paragraphs for manual and synthetic data).

The main source of potential bias is the number of added tokens in the edits, which trends much higher for synthetic data than manual data (158 vs 89 tokens for manual vs synthetic). This is expected to some degree, due to the minimum matching length used in our synthetic labeling algorithm. This length bias leads to an increased average unigram overlap between review comments and edits, which is higher for synthetic data (8.3%) than for manually-annotated data (6.6%). However, when we control for length by binning comment-edit pairs based on the geometric mean of comment and edit length, the average difference between bins is only 0.4%.

## C   GPT-4 PROMPTS

Here we provide the prompts used for the GPT experiments. Because it was only feasible to evaluate the pairwise GPT methods on fully-additive edits, note that we designed the prompts accordingly.

The prompts were created with 10-20 iterations of manual adjustment on a small handful of instances from the training set. We also found in preliminary experiments that the 1-shot GPT pairwise setting did better when the negative example was given first and the positive example second; we suspect that the model is biased towards an alternating `[yes, no, yes, no]` sequence of examples, and because the majority of candidates are negatives it is better to set up the sequence to bias in favor of a "no".

### C.1  GPT-PAIRWISE (0-SHOT)

Consider the following review comment for a scientific paper: **<comment>**

Consider the following paragraph, which was added to the paper after the review:
**<edit>**

Is the new paragraph likely to have been added for the purpose of addressing this review comment? Answer with "yes" or "no".

### C.2  GPT-PAIRWISE (1-SHOT)

You need to determine which edits correspond to a given reviewer comment for a scientific paper. Given a comment and a paper edit (where changes are enclosed by brackets with +/- to indicate additions/deletions), you must determine whether the edit was likely added to the paper to address the comment. Here are some examples:
**<examples, formatted identically to the main query below, followed by "Answer: yes|no">**

Consider the following review comment for a scientific paper: **<comment>**

Consider the following paragraph, which was added to the paper after the review:
**<edit>**

Is the new paragraph likely to have been added for the purpose of addressing this review comment? Answer with "yes" or "no".

### C.3  GPT MULTI-EDIT

Consider the following comments that a reviewer made about a scientific paper (each followed by a unique comment id):

**<comment>**
comment id: **<comment id>**

**<repeat for all comments>**
Below is a partial diff of the original paper text and the paper text after the authors made revisions in response to various reviews. Changes are indicated with brackets "[]" with a "+" for additions and a "-" for deletions. Below each paragraph is a unique "edit id". Determine which edits were meant to address the given reviewer comments above.
—BEGIN PAPER DIFF—
**<edit>**
section: **<section name>**
edit id: **<edit id>**

**<repeat above until out of edits or reached model token limit>**
—END PAPER DIFF—

Which edit ids correspond to each of the reviewer's comments? The relationship is many-to-many; one comment could correspond to several edits, and several comment could correspond to the same edit. There could also be comments that the authors didn't address at all or edits that were not made in response to any particular comment.

Write the answer as JSON lines with the format {"comment_id": <comment id>, "edit_ids": [<edit ids>], "notes": ""} where each record has a comment id and the list of edit ids that correspond to it. The "notes" field is optional and can contain any notes about edits you weren't sure about or reasons for including/omitting certain edits.

While the "notes" field in the prompt was included in the hope that it might provide insight into the model's reasoning and assist in diagnosing its errors, in practice we found that the notes were usually not very informative (e.g., *"this comment was not addressed by the edits"*), and therefore we did not do a formal analysis of the model's notes.

### C.4 GPT EDIT GENERATION

Consider the following excerpt of a scientific paper which is under review for a conference:

— START —
Abstract: **<abstract>**

Body: **<sequence of body paragraphs>**
— END —

A reviewer made the following comment about the paper: **<comment>**

Write a response to the reviewer and an edit (or edits) that could be added somewhere in the paper (or Appendix) to resolve the reviewer's comment. Above an edit, write the location in the paper where it should be made. The edit should not explicitly say that it is written in response to a reviewer comment; it just needs to improve the paper such that a future reviewer would be unlikely to make the same comment. If addressing the comment requires additional experiments or information that you do not have access to, you can use placeholders or fill in reasonable guesses for that information. An edit may be a new sentence, paragraph, or section, depending on the comment.

For ease of parsing, write "Response:" before the reviewer response, "Location:" before the edit location(s), and "Edit:" before the edit(s).

The above prompt asks for an author response, which we did not discuss in the main paper. During preliminary prompt tuning, we found that the model had a tendency to phrase its edits as though it was writing a response directly to the reviewer (often including a phrase along the lines of, "as the reviewer suggests, we ..."). Encouraging the model to write a direct response separate from the paper edit appeared to mitigate this tendency and improve the quality of the edits.

## D ADDITIONAL ANNOTATION INFORMATION

To extract actionable comments from reviews, annotators were shown the review text in a web-based annotation interface where they could highlight spans corresponding to comments. The annotators were instructed to select comments based on the definition in subsection A.3. That is, any comments which imply some action should be performed to improve the paper are included, but non-actionable comments such as summaries, positive comments, or comments too vague and fundamental to be

addressable (*"Lacks novelty."*) are excluded. As a rule of thumb for unclear cases, a comment was included if the annotators could imagine rewriting it as an imperative to-do list item.

The interface allowed for selecting arbitrary token spans, but in practice almost all spans aligned roughly to sentence boundaries. The majority (78%) of extracted comments were one sentence long and some (19%) were two sentences long, with only 4% being more than two sentences.

Table 7: Statistics for manually- and synthetically-labeled data. Papers, reviews, and aligned edits are counted only when they correspond to included comments. Edits are counted only once, even if they correspond to multiple comments.

| Statistic | Manual | Synthetic |
|---|---|---|
| Papers | 42 | 1678 |
| Comments | 196 | 3892 |
| Aligned Edits | 131 | 3184 |

# E  IMPLEMENTATION DETAILS

For training models on the comment-edit alignment task, we sample 20 negative edits for each comment. The negative edits are sampled from the pool of training documents excluding the one to which the comment applies, to mitigate the low recall of synthetic data.

For all methods, we exclude edited paragraphs with fewer than 100 characters, as shorter ones are often badly parsed equations or text fragments that appear erroneously as edits.

All neural models are trained on NVIDIA Quadro RTX 8000 and RTX A6000 GPUs. We use the Adam optimizer (Kingma and Ba, 2015) with a learning rate of 2e-5, batch size of 16, and $\beta$ of [0.9, 0.999], running for a maximum of 8192 steps and selecting the best model on the dev set. The experiments are implemented using Pytorch 1.10 (Paszke et al., 2019) and Huggingface Transformers 4.21 (Wolf et al., 2020) for transformer models and Gensim 4.3 (Řehůřek and Sojka, 2010) for BM25. For GPT-4, we use the `gpt-4-0314` model and use a temperature of zero in all experiments.

## E.1  ALIGNMENT MODEL DETAILS

**Bi-encoder:**  The model separately consumes each review comment and edit to create an embedding for each, with a goal that embeddings for corresponding comments and edits are closer to each other than those for non-corresponding pairs are. We prefix the comments with "review comment:" and the edits with "edit:" to allow the model to treat the two text types differently. For fine-tuning, we use a triplet loss; given a triplet consisting of a comment $c$, a positive edit $x_+$, a negative edit $x_-$, and a cosine similarity function $\text{sim}(\cdot, \cdot)$, the loss is

$$\mathcal{L} = \max(0, \text{sim}(c, x_-) - \text{sim}(c, x_+) + 0.5)$$

The bi-encoder models are DeBERTaV3-large (He et al., 2021) and SPECTER2 (Singh et al., 2022).

**Pairwise cross-encoder:**  The model consumes a comment-edit pair separated by a `[SEP]` token and outputs a score representing the likelihood of a positive label. DeBERTaV3-large (He et al., 2021), LinkBERT (Yasunaga et al., 2022), and GPT-4 (OpenAI, 2023) models are used with this format. For GPT-4, we try both a zero-shot setting where only instructions are given and a (2-way) one-shot setting where one positive and one negative example are given in the prompt.

**Multi-edit cross-encoder:**  The model consumes all edits for a paper at once, including unchanged paragraphs as "edits" for context; in essence, this is a full "diff" of the paper with an edit ID number attached to each paragraph. We additionally feed all comments for a paper at once, each with a unique ID. The output is formatted as a list of JSON objects, each containing a comment ID and a list of edit IDs. In practice, a diff of the full paper is often too long to fit model length limitations, and in these

cases we split the paper into 2-3 chunks and merge the output lists. We use GPT-4 (OpenAI, 2023) for this variant.[4]

**Bag of words:**   We try a simple BM25 ranker (Robertson and Zaragoza, 2009) that scores a comment against the post-revision text of an edit.

For all models that produce similarity scores or probability outputs, we tune a decision threshold on the dev set to maximize micro-F1. In addition, we use a version of BM25 tuned for high recall (>90%) on the dev set as a first-pass candidate filter for the GPT-4 based methods, which increases evaluation speed and reduces GPT-4 API costs.

## F   MACRO-F1 BIAS

The macro-F1 scores we report in Table 1 show different relative performance between some models compared to the micro-F1 results. Notably, BM25 does very well on macro-AO-F1 and the DeBERTa cross-encoder does much worse than the DeBERTa bi-encoder on macro-F1 (whereas the results were comparable with micro-F1). These differences can be explained as the result of a bias in macro-F1 that favors more conservative models (those that output fewer positive predictions) on our data.

The bias stems from the fact that some papers, the comments for the paper do not have any corresponding edits (i.e., when authors did not address the comments), and if a model correctly predicts zero alignments in these cases, we consider the F1 as 100 (the F1 is technically undefined in this case). Because the average F1 scores are somewhat low, getting a few perfect 100 scores can skew the results (and outputting even a single positive prediction in those cases changes the result to 0), and thus a model that is biased to output fewer positives has an advantage.

We also note that some comments are associated with multiple edits. Less conservative models might have an advantage in such cases by achieving higher recall, but when macro-averaging the groups with many positive labels are effectively downweighted, mitigating the potential disadvantages of more conservative models.

As an example of the bias, consider the DeBERTa cross-encoder compared to the DeBERTa bi-encoder. The cross-encoder models tend to output tighter score distributions and are thus more susceptible to inconsistencies between the dev and test sets; in this case the DeBERTa cross-encoder outputs 17% positive predictions on the test set pairs while the bi-encoder outputs only 3% (the true percentage is 1.7%). This interacts with the bias of macro-F1 to favor conservative models and results in a large difference in performance. The DeBERTa bi-encoder model gets perfect scores on an extra 12% of the groups compared to the cross-encoder, which more than compensates for the fact that it typically does worse on the groups where there are positive labels.

## G   COMMENT-SOURCE ALIGNMENT

While comment-edit alignment maps a comment to a corresponding edit made to the paper, *comment-source alignment* maps a comment to the corresponding source paragraph where an edit should be made. Good comment-source alignment would make it possible to break down the edit generation task into separate location-finding and edit-generation stages, which could improve efficiency and accuracy.

Comment-source alignment is more challenging than comment-edit alignment because it doesn't provide information about the content of the edit. It is also more ambiguous: there are sometimes multiple places where an edit could potentially be made to address a comment, and we only use the location of the real edit as a correct answer. Nonetheless, in this section we apply our baseline models to the task to quantify its difficulty.

We use the same baseline models as in section 5, excluding GPT-4 due to its high cost. To reduce the degree of ambiguity in the task, we reduce our dataset to include only comment-edit pairs that correspond to specific source paragraphs (excluding newly-added paragraphs).

---

[4]OpenAI has indicated plans for a 32k-sized model, but that is not available to us as of this work.

## G.1 RESULTS

Results are reported in Table 8. While results are not directly comparable to comment-edit alignment due to the different dataset split, the micro-f1 results are all substantially lower; macro scores are higher, but this is likely the result of models being biased towards predicting negatives and therefore getting more perfect scores on comments that have no aligned edits.

Table 8: Precision (P), Recall (R), and F1 of comment-source alignment on test data. The micro-average is over all comment-edit pairs, while the macro-average is grouped by paper.

| Model | Micro | | | Macro | | |
|---|---|---|---|---|---|---|
| | P | R | F1 | P | R | F1 |
| BM25 | 5.0 | 3.5 | 4.1 | 57.9 | 43.7 | 26.2 |
| **Bi-encoder** | | | | | | |
| Specter2 no-finetune | 3.5 | 22.8 | 6.1 | 15.1 | 54.4 | 16.7 |
| Specter2 | 3.0 | 13.5 | 4.8 | 24.6 | 45.8 | 14.8 |
| DeBERTa | 0.4 | 0.6 | 0.5 | 51.0 | 40.6 | 22.4 |
| **Cross-encoder** | | | | | | |
| LinkBERT | 1.4 | 0.6 | 0.8 | 54.9 | 40.9 | 22.4 |
| DeBERTa | 2.3 | 7.6 | 3.0 | 39.6 | 43.8 | 16.9 |

