# OpenReview forum: "ARIES: A Corpus of Scientific Paper Edits Made in Response to Peer Reviews"
_ICLR.cc/2024/Conference — Submitted to ICLR 2024_

### Official Review · Reviewer_md3r · 2023-10-31

**Soundness:** 4 excellent
**Presentation:** 4 excellent
**Contribution:** 3 good
**Rating:** 8
**Confidence:** 3

**Summary:**

This presents a dataset of paper review comments aligned to paper edits. It extracts an "edits" set somewhat cleverly by just getting different paper versions from openreview, doing pdf-parsing, and looking at differences, and then compares those changes to comments in reviews of the paper to detect when changes seem to be entailed by a specific change in the paper.   They provided a small dataset of full manual alignment between those edits and the reviews and a larger low-recall high-precision automatically evaluated dataset using an additional signal from rebuttals. Models are then presented both for generating those edits in response to review comments and for locating them within a text.

**Strengths:**

- The biggest strength is that this paper provides an extensive analysis of the contours of their task and analysis of the underlying phenomena.
-They treat the task with relevant amounts of delicacy, by being very explicit about models lacking the lab notes/data to make many of these edits and doing labeling of actionability to study the issue. This adds nuance to what might otherwise be a problematic task with factuality issues.
- The model work on comment-edit alignment seems relatively rigorous, using domain-relevant models like SPECTRE2.

**Weaknesses:**

1) The core manual data is very tiny (196 alignments, 42 reviews total) and so some of the value of the dataset really rests on the quality of the synthetic alignment work detailed in A.5.
2) It's hard to have a good intuition about the quality of their IAA with Jaccard overlap of 65% in the reviews, but it does call into question whether the segmentation/identification of relevant review comments is clear.  Insofar as there is the whole field of peer review segmentation and typing  (e.g. Xua et al. 2021, Cheng et al. 2021, Kennard et al. 2021, Dycke et al. 2023), it might be worth checking if any of that is relevant for use (some of those may also be relevant to the analysis types).

**Questions:**

I appreciated the "action class" analysis the authors provided over the manual data, and am curious whether there is data (or even impressions) regarding whether that distribution of types is actually the same in the synthetic data. Wouldn't some types of actions be more likely to have their corresponding edits repeated in the rebuttal (e.g., explain) and some types very unlikely to be repeated in the rebuttal (e.g., remove)?
If this is envisioned as part of a workflow, does it really make sense for the model to be freely determining in effect whether to comply, disagree, make promises, etc.?  Wouldn't those components ideally make sense as a starting assumption for an edit?

---

> ### Author Response · Authors · 2023-11-22
>
> Thank you for your review; here are some responses to your comments:
>
> > The core manual data is very tiny (196 alignments, 42 reviews total) and so some of the value of the dataset really rests on the quality of the synthetic alignment work detailed in A.5.
>
> It is true that the number of comments and reviews is relatively small, although we would note that we actually have over 10k comment-edit candidate pairs in our alignment experiments, and for edit generation our analysis produced statistically significant conclusions despite the small number of annotations.
>
>
> > It's hard to have a good intuition about the quality of their IAA with Jaccard overlap of 65% in the reviews, but it does call into question whether the segmentation/identification of relevant review comments is clear.  Insofar as there is the whole field of peer review segmentation and typing (e.g. Xua et al. 2021, Cheng et al. 2021, Kennard et al. 2021, Dycke et al. 2023), it might be worth checking if any of that is relevant for use (some of those may also be relevant to the analysis types).
>
> Of note, we did find that 88% of spans overlap between annotators; the reason for token-level jaccard being lower is that different amounts of context were selected (but qualitatively the overall meanings were similar).
>
> Many of those prior works don't seem to report agreement for segmentation, but for those that do, they also find a fair amount of noise in the annotation.  While not perfectly apples-to-apples with our metric, Kennard et al (2022, DISAPERE) reports Cohen's kappa in the range of 0.5-0.6 for action/aspect annotation (their Table 5), while Hua et al (2019, AMPERE) reports Krippendorf's alpha of 0.61 for type annotation and Cohen's kappa of 0.93 for proposition segmentation (their page 2).  Our comment extraction consists of identifying "actionable" comments, so it can be thought of as a combination of segmentation and typing in that regard.
>
>
> > I appreciated the "action class" analysis the authors provided over the manual data, and am curious whether there is data (or even impressions) regarding whether that distribution of types is actually the same in the synthetic data. Wouldn't some types of actions be more likely to have their corresponding edits repeated in the rebuttal (e.g., explain) and some types very unlikely to be repeated in the rebuttal (e.g., remove)?
>
> Indeed, while we have not done a formal analysis of classes for the synthetic data, we can say anecdotally that we did not see any "remove" type comments in the sample of 50 that we checked when validating the accuracy of the synthetic data.  No other significant differences stood out, however.
>
> Out of curiosity, we did an exploratory analysis using GPT-4 to label the examples.  It is not entirely accurate, reaching only 74% accuracy on the manually labeled data, but the overall distribution is still similar to the one we obtained from manual annotation.  The table below shows the GPT-scored frequencies of both the manual data and the synthetic data (dev set).
>
> ```
> Action     | Manual | Synthetic
> explain    |    44% |       48%
> compare    |    16% |       19%
> report     |    12% |       16%
> use        |     7% |        7%
> discuss    |     7% |        4%
> apply      |     5% |        5%
> remove     |     4% |        1%
> define     |     4% |        1%
> ```
>
> From the table, it appears that the distributions are fairly similar, and given the uncertain fidelity of GPT it may be unwise to put much weight on the differences.  Nonetheless, the synthetic data does appear to have slightly more instances of asking to explain/compare/report and fewer instances of asking to remove/define/discuss.
>
>
> > If this is envisioned as part of a workflow, does it really make sense for the model to be freely determining in effect whether to comply, disagree, make promises, etc.? Wouldn't those components ideally make sense as a starting assumption for an edit?
>
> This depends on the result to some extent.  Given our result that the model does not match humans very well when writing edits, it would seem that in practice a user should provide some guidance to start the edit.  On the other hand, if models improve or if we can reliably determine which comments a model can address well, then it might make sense to let the model do some work without user input.
>
> Another factor to consider is that it takes some work for an author to provide guidance on each edit, so it may be easier to receive a candidate edit, decide whether to accept/reject, and potentially revise the edit as needed, even if this results in more generated edits being rejected by the user.  The accept/reject approach has been used in some past work, but more user studies will be needed to compare the different ways of integrating these assistants into practical workflows.

---

### Official Review · Reviewer_Lkvy · 2023-11-01

**Soundness:** 2 fair
**Presentation:** 2 fair
**Contribution:** 2 fair
**Rating:** 3
**Confidence:** 4

**Summary:**

This paper presents newly constructed data of review comments and their corresponding edit parts in the original and revised versions of scientific papers taken from OpenReview. They introduced two tasks related to this corpus data: The comment-edit alignment task is to identify the correspondence between a comment and an edit, and the edit generation task is to generate an edit given a reviewer's comment.
For the former task, they applied a number of binary classification models, evaluated the results, and discussed the causes of errors. For the latter task, they applied GPT-4 to generate edit responses given a comment, and evaluated and discussed the difference between human and GPT-4 generated edit responses.

**Strengths:**

- The paper provides unique data that includes reviewers' comments on scientific papers and their corresponding edits obtained from the original and revised versions of the papers.
- They applied a recent generative language model, GPT-4, to tackle two problems they defined, comment-edit alignment and edit generation tasks, and gave a detailed analysis of the results.

**Weaknesses:**

- The motivation and the usefulness of the proposed tasks are not clear. For the first task, comment-edit alignment, there are a number of papers that require the authors' answer letter to the editors that describe how the authors have responded to each of the comments raised by the reviewers. For such journal papers, there is no need to find correspondence between the edited parts of the revision. The second task is more puzzling. As the authors claim, GPT-4 often addresses responses on a superficial level without including technical details. It is totally unclear why such superficial or pretending responses are necessary.
- Even though the comment-edit alignment task is a binary classification task, the overall results are very low, much lower than 50%, which may imply that this data or this task is an ill-formed one.
- The details of the data construction and the task description are not understandable without referring to appendices.

**Questions:**

- Why are the results of micro scores for the comment-edit alignment so low, even though it is a binary classification task? What is the proportion between the positive and negative pairs, and what scores will be obtained under random guess?
- In the macro evaluation, what is the reason that F1 scores are lower than both precision and recall? How are those F1 scores calculated?
- According to the precision and recall scores for the macro evaluation, the performance of BM25 looks better than or at least competitive with those of GPT-4 multi-edit. It would be better to discuss the differences between the results of those models.
- What is the motivation behind setting the edit generation task? This looks to be an unanswerable question by other than the authors. The motivation as well as the usefulness of the task should be described.

**Details Of Ethics Concerns:**

The paper is not quite related to representation learning.

---

> ### Author Response · Authors · 2023-11-22
>
> Thank you for your review; here are some responses to your comments (split into two parts due to the character limit):
>
> > For the first task, comment-edit alignment, there are a number of papers that require the authors' answer letter to the editors that describe how the authors have responded to each of the comments raised by the reviewers. For such journal papers, there is no need to find correspondence between the edited parts of the revision.
>
> The purpose of the comment-edit alignment task is threefold: (1) it provides a challenging reasoning task to test the capabilities of LLMs, (2) if solved, it would provide a way to automatically obtain large-scale aligned data, which could be used to study peer reviewing and to train edit- and review-generation models, and (3) because many venues do not require authors to describe the edits or do not enforce those rules, automatic alignment could still be useful in tools that assist with tracking changes during peer review discussions.
>
>
> > The second task is more puzzling. As the authors claim, GPT-4 often addresses responses on a superficial level without including technical details. It is totally unclear why such superficial or pretending responses are necessary.
>
> Similar to comment-edit alignment, the edit generation task is motivated as both a challenging reasoning task and as a potentially useful task in writing-assistance applications.  Of course, we would not expect GPT to generate perfect edits due to both reasoning limitations and data limitations, but it is not clear in advance how well it could do.
>
> One might imagine that GPT could produce edits that are very realistic, perhaps with some placeholders for missing information, and that an author could simply make some minor adjustments to update the paper.  However, what we find in this work is that the kinds of edits GPT generates differ from the kinds of edits that humans generate in several significant ways, such as having a high tendency to paraphrase the comment being addressed (which human authors rarely do).
>
> We hope that our findings and dataset can motivate and assist future work on improved edit generation techniques.
>
>
> > Even though the comment-edit alignment task is a binary classification task, the overall results are very low, much lower than 50%, which may imply that this data or this task is an ill-formed one.
>
> > Why are the results of micro scores for the comment-edit alignment so low, even though it is a binary classification task? What is the proportion between the positive and negative pairs, and what scores will be obtained under random guess?
>
> The scores are below 50% because it is not a balanced binary classification task; the number of positives is quite small (1.7%) because most comments do not correspond to most edits (and vice versa); in general we would expect that with N comments and M edits per paper, the fraction of positive-labeled pairs would be proportional to 1/(N*M).
>
> We have computed the scores of a random-guess classifier (using the test proportion of positive examples as the positive-guess probability) and added the results to Table 1.  All methods outperform random on almost all metrics, with the exception of deberta-biencoder and linkbert-crossencoder on micro-ao-f1; these have a fairly tight distribution of probability outputs and the threshold chosen on the dev set was a poor match for the test set.

---

> ### Author Response · Authors · 2023-11-22
>
> > In the macro evaluation, what is the reason that F1 scores are lower than both precision and recall? How are those F1 scores calculated?
>
> The F1 is lower than P/R because the harmonic mean is lower than the arithmetic mean, and we are doing an arithmetic-mean macro average.  That is, we are individually computing the P/R/F1 for each group of comment-edit pairs (grouped by comment originally, and now by paper; see below for details) and taking the arithmetic mean of those to get the macro-F1.  We could imagine an extreme case where half of those groups have P=10, R=100, F1=18 and the other half have P=100, R=10, F1=18 and the macro-averages are ultimately P=55, R=55, F1=18.
>
>
> > According to the precision and recall scores for the macro evaluation, the performance of BM25 looks better than or at least competitive with those of GPT-4 multi-edit. It would be better to discuss the differences between the results of those models.
>
> The high performance of BM25 is largely due to a bias in macro-f1, which favors methods that are conservative in their predictions (i.e., tending to output fewer positive predictions).  We agree that this makes it somewhat confusing to interpret, so we have changed the macro-average to group by paper instead of by comment, which reduces the effect of the bias.  For completeness, we explain the bias below:
>
> The macro-f1 bias in question stems from the fact that some comments do not have any corresponding edits (i.e., when authors did not address a comment), and if a model correctly predicts zero alignments in these cases, we consider the F1 as 100% (the F1 is technically undefined in this case).  Getting a few perfect 100 scores can skew the results, and thus a model that is biased to output fewer positives has an advantage.  This is exacerbated by the fact that some comments correspond to multiple edits; models biased to output more positive predictions are more likely to do well when there are several positive labels, and yet these cases are effectively down-weighted by macro-averaging.  Thus, while GPT is much better at identifying any given comment-edit pair (as indicated by the micro-average), the macro average has the effect of up-weighting its false positives and down-weighting its true positives.
>
> We have added some discussion of this bias in Appendix F of the paper.

---

### Official Review · Reviewer_nrwT · 2023-11-02

**Soundness:** 3 good
**Presentation:** 3 good
**Contribution:** 4 excellent
**Rating:** 8
**Confidence:** 4

**Summary:**

The paper introduces the challenging task of revising scientific papers based on peer feedback. The authors present ARIES, a valuable dataset of review comments and corresponding paper edits, to facilitate training and evaluation of models. The study focuses on two subtasks: comment-edit alignment and edit generation. Experiments reveal that existing models, including GPT-4, struggle with these tasks. While GPT-4 can generate edits on a surface level, it rigidly follows the wording of feedback rather than capturing the underlying intent and tends to include fewer technical details than human-written edits. The findings suggest the need for further research in this area, emphasizing the complexities of reasoning about scientific text and addressing challenges in aligning feedback to edits and generating meaningful revisions.

**Strengths:**

+ Automatically revising scientific papers based on peer feedback is a meaningful task. Decomposing the task into comment-edit alignment and edit generation is well-motivated.

+ The constructed ARIES dataset has its great practical values in improving NLP systems for not only scientific paper revision tasks but also other tasks during the peer review process.

+ The empirical analyses are comprehensive, with meaningful case studies and error analyses. Different model architectures (BM25, BERT, GPT-4; bi-encoder, cross-encoder) are examined. In particular, the observations of GPT-4's performance on edit generation are inspiring.

**Weaknesses:**

- The technical novelty is somehow limited, although I understand that the main contribution of this submission is on the dataset and benchmark. All compared approaches are existing models. After the authors obtain meaningful observations from empirical studies, they do not further design an effective method based on their observations to achieve better performance.

- This may be a common criticism for any paper showing that LLMs do not perform well on a certain task: It is possible that the poor performance of GPT-4 is due to inappropriate instructions or prompts. More analyses are needed on the effect of instructions. For example, if chain-of-thoughts prompting is used, would GPT-4 generate "deeper" edits?

**Questions:**

- Could you try chain-of-thoughts prompting or other more advanced techniques to see whether the performance of GPT-4 can be improved?

- I would suggest directly writing "SPECTER2" rather than "SPECTER" in Table 1. Also, could you specify which adapter is used for SPECTER2?

- The following reference may be very relevant to this paper, considering using GPT-4 for writing peer reviews.

[1] Can large language models provide useful feedback on research papers? A large-scale empirical analysis. arXiv 2023.

---

> ### Author Response · Authors · 2023-11-22
>
> Thank you for your review; here are some responses to your comments:
>
> > It is possible that the poor performance of GPT-4 is due to inappropriate instructions or prompts. More analyses are needed on the effect of instructions. For example, if chain-of-thoughts prompting is used, would GPT-4 generate "deeper" edits?
> > Could you try chain-of-thoughts prompting or other more advanced techniques to see whether the performance of GPT-4 can be improved?
>
> A proper chain-of-thought analysis may be difficult to do in the given time, but we did spend ~20 manual iterations to find a good prompt (using train examples to test).  Since the design space is quite large for this task, we limited scope to single-pass prompts, but in future work we would like to explore more options to improve GPT's ability to work with technical text, such as fine-tuning, multi-persona self-discussion, chain-of-thought, and perhaps user intervention.
>
>
> > I would suggest directly writing "SPECTER2" rather than "SPECTER" in Table 1. Also, could you specify which adapter is used for SPECTER2?
>
> Thank you for pointing this out, we have updated the paper to specify the model.  In preliminary experiments we tried the PRX adapter and the base model (with no adapter) and found that the base model was better on the dev set, so that is the one we report.
>
>
> > The following reference may be very relevant to this paper, considering using GPT-4 for writing peer reviews.
>
> Thanks for mentioning this, it is very interesting.  One future direction that we didn't have space to discuss is the possibility of linking review generation and edit generation to create chain-of-thought style automatic paper editing with LLMs.
>
> Although this is slightly outside the scope of the current work, you might be interested in this finding: for a different project, we evaluated the review generation approach from that work (and other methods) and found a similar trend to what we observed with edit generation in this work.  Namely, the comments from GPT were generally not unreasonable but tended to be generic and surface-level things that you could say about many papers, e.g., `The paper might not provide enough details about the implementation of $METHOD. Evaluate whether the paper provides sufficient details about $METHOD for it to be implemented by others.`  It generally doesn't give detailed suggestions or provide a clear rationale for why the suggestions are important.  But, we also found hope for improving specificity through internal self-debate and careful prompting, albeit at a much higher computational cost.

---

### Official Review · Reviewer_fRaq · 2023-11-03

**Soundness:** 2 fair
**Presentation:** 2 fair
**Contribution:** 2 fair
**Rating:** 3
**Confidence:** 4

**Summary:**

This paper introduces a new data set named ARIES with scientific reviews, where each comment is matched to a paper edit. This data set could be used to detect which edits correspond to the request and for an edit generation from comment task. The data set consists of 196 human annotations across 42 reviews, judgements if statements are actionable and 3900 comments automatically matched with high precision and low recall. The paper present several approaches to align comments to edits and studies GPT-4 for generating the edits.

**Strengths:**

Good annotation task setup and agreement metrics obtained on these annotations.

**Weaknesses:**

The data set that is manually annotated is small (196 annotations across 42 reviews) and may not be diverse enough.

The generation task setup is very challenging and includes multiple aspect that would not be available to a predictive model: the context of research at that point, historical information that may be relevant or the goals of the authors which for example would not like to admit a weakness in their publication. Because of this, I think potential applications and extensions would be important to mention.

Experiments in Section 5 require more information about how the models were trained, especially how the negatives are selected. The results currently show better results on Macro F1 for the non fine-tuned model, and worse results when using cross-encoders as compared to bi-encoders, which is unintuitive and looks quite suspicious. Another detail that needs to be mentioned are how the units of text that are matched are computed (a sentence, a paragraph?)

There are several key assumption made in the data set construction that I think should be better highlighted or organized, such as that responses can be presented in the forum rather than in edits to the paper.

The experiments in section 6 are only performed using an off the shelf GPT model.

**Questions:**

NA

---

> ### Author Response · Authors · 2023-11-22
>
> Thank you for your review; here are some responses to your comments (split into two parts due to the character limit):
>
> > The data set that is manually annotated is small (196 annotations across 42 reviews) and may not be diverse enough.
>
> It is true that the number of comments and reviews is relatively small, although we would note that we actually have over 10k comment-edit candidate pairs in our alignment experiments, and for edit generation our analysis produced statistically significant conclusions despite the small number of annotations.
>
>
> > The generation task setup is very challenging and includes multiple aspect that would not be available to a predictive model: the context of research at that point, historical information that may be relevant or the goals of the authors which for example would not like to admit a weakness in their publication. Because of this, I think potential applications and extensions would be important to mention.
>
> Thank you for this suggestion; it would be interesting to see how edit generation can improve by incorporating retrieval, multimodal information, and user interaction, and this is something we would like to explore in future work.  Nonetheless, we argue that the simple GPT-based setup is still useful to study and should be able to answer many instances without additional context, as it has a large amount of scientific background knowledge from pretraining.  In our answerability analysis (Table 4), we found that about half of the comments should be answerable with just the given paper and strong background knowledge.
>
> Also of note is the fact that a single paper often exceeds 8k tokens (the highest limit available to us at the time) and in rare cases even exceeds 32k tokens, so there is a tradeoff between how much related information could be included vs how much context from the paper being edited.
>
>
> > Experiments in Section 5 require more information about how the models were trained, especially how the negatives are selected.
>
> We provide some information about negative sampling at the start of section 5 and in Appendix E; are there specific details you were interested in?  To clarify: For manually-annotated data (the test data, which our numbers are reported on), for a given comment, we consider all edits for the corresponding paper as candidate edits, labeled as positive if the edit was annotated as addressing the comment and negative otherwise.  So the negatives are all non-aligned edits for the same paper.
>
> For training we use the synthetic data.  Given the low recall of the synthetic data (discussed in subsection A.5), we can only use the synthetic labels to produce positive comment-edit alignment pairs; thus, we pair comments with 20 edits sampled from other documents as negative candidates.
>
> Appendix E also provides some more training details, such as hyperparameters and hardware details.
>
>
>
> > Another detail that needs to be mentioned are how the units of text that are matched are computed (a sentence, a paragraph?)
>
> > There are several key assumption made in the data set construction that I think should be better highlighted or organized, such as that responses can be presented in the forum rather than in edits to the paper.
>
> Thanks for the suggestion; we have added some clarification to section 4 about the units (edits are paragraph-level, comments are technically token spans but in practice are almost always sentences) and the many-many nature of the mapping, and we tried to make the pointer to the appendix more prominent since we didn't have space for the full construction details.
>
>
> > The experiments in section 6 are only performed using an off the shelf GPT model.
>
> We agree that this is a limitation, and in future work we hope to develop better methods for edit generation.  In this work our goal was to release the dataset and to study the behavior of existing methods on our tasks as a baseline, and focusing on just one method for edit generation allowed us to present a more thorough analysis.

---

> ### Author Response · Authors · 2023-11-22
>
> > The results currently show better results on Macro F1 for the non fine-tuned model, and worse results when using cross-encoders as compared to bi-encoders, which is unintuitive and looks quite suspicious.
>
> The results stem in part from a bias in the macro-f1 metric that favors more conservative models (that is, those that output fewer positive labels), which we explain in more detail below.  We believe the micro-f1 scores are more reliable and easier to interpret, and we note that on the micro-f1 scores, the finetuned model is much better for Specter2.  We also find that for the same (DeBERTa) model, the cross-encoder and bi-encoder setups perform somewhat similarly on micro-f1 (and the cross-encoder does better on micro-ao-f1).
>
> We had included the macro-f1 scores in an attempt to account for the possibility of some comments being especially error-prone (e.g., if one comment resulted in many false positives for some reason, it could skew the micro-averaged results).  However, we agree that the macro scores are not intuitive to interpret, so we have changed them to group by paper instead of by comment, which reduces (but does not entirely eliminate) the bias.
>
> In the new Table 1, the finetuned results are much better than non-finetuned, as expected.  The cross-encoder results still underperform the bi-encoder results.  The cross-encoder models tend to output tighter score distributions and are thus more susceptible to inconsistencies between the dev and test sets; in this case Deberta-cross outputs 17% positive predictions on the test set pairs while Deberta-bi outputs only 3% (the true percentage is 1.7%).  This interacts with the bias of macro-f1 to favor conservative models on our data and results in worse performance.
>
> The macro-f1 bias in question stems from the fact that some comments do not have any corresponding edits (i.e., when authors did not address a comment), and if a model correctly predicts zero alignments in these cases, we consider the F1 as 100 (the F1 is technically undefined in this case).  Because the average f1 scores are somewhat low, getting a few perfect 100 scores can skew the results (and outputting even a single positive prediction in those cases changes the result to 0), and thus a model that is biased to output fewer positives has an advantage.  For example, the non-finetuned Specter2 model gets perfect scores on an extra 6% of groups compared to the finetuned one, which compensates for the fact that it does worse on the groups where there are positive labels.  By changing the grouping to be per-paper instead of per-comment, we reduce the number of groups with no positive labels.
>
> We also note that some comments are associated with multiple edits.  Less conservative models might have an advantage in such cases by achieving higher recall, but when macro-averaging the groups with many positive labels are effectively downweighted, mitigating the potential disadvantages of more conservative models.
>
> We have added some discussion of this bias in Appendix F of the paper.

---

### Meta-Review · Area_Chair_ocL9 · 2023-12-07

**Metareview:**

This paper introduced the novel tasks of comment-edit alignment and edit generation for scientific paper revisions based on high-level draft feedback from reviewers. It constructs a dataset called ARIES for evaluating the tasks. Several baselines, including GPT-4, are evaluated and compared on this dataset, and the results are presented and discussed.

Strengths: The comment-edit alignment and edit generation tasks proposed in this paper are new and the constructed dataset might be useful for future researches. Detailed experimental analysis is provided.

Weaknesses:  The manually annotated dataset is small. The technical novelty of dataset construction and the methods/models used to address the two tasks is limited. There is no specific method design to overcome the challenges of the proposed tasks. More analyses are needed on the effect of instructions for LLMs.

**Justification For Why Not Higher Score:**

see the meta-review.

**Justification For Why Not Lower Score:**

N/A

---

### Decision · Program_Chairs · 2024-01-16

Reject